# Qualitative exploration of comprehension and experiences of healthcare professionals regarding nutrition care in Karachi, Pakistan

**Naeema Tahira Syed**[1], **Sukaina Shabbir**[2], **Maria Atif**[3]*, **Rubina Hakeem**[4]

**1** Faculty of Allied Health Sciences, College of Human Nutrition and Dietetics, Ziauddin University, Karachi, Sindh, Pakistan, **2** School of Public Health, Dow University of Health Sciences, Karachi, Sindh, Pakistan, **3** School of Public Health, Dow University of Health Sciences, Karachi, Sindh, Pakistan, **4** Nutrition Department, LiveGreen Academy, New Haven, Connecticut, United States of America

* maria.atif@duhs.edu.pk, drmariatif@yahoo.com

## Abstract

Nutrition care involves providing safe and adequate food and beverages through a systematic, evidence-based approach aimed at enhancing or restoring the nutritional well-being of individuals or populations. In Pakistan, due to structural and economic limitations, doctors and nurses often deliver nutritional care instead of trained dietitians or nutritionists. This study was conducted to qualitatively explore the understanding and experiences of these healthcare providers in delivering nutrition care to a sample of the urban Pakistani population. The study included 15 healthcare professionals working in various hospital settings. Semi-structured interviews were conducted to investigate participants' understanding, current practices, challenges, and suggestions for providing nutrition care in Pakistan. A purposive sampling method was used to enroll participants. Following informed consent, data were audio-recorded, transcribed verbatim, anonymized, and translated into English. Thematic analysis was performed using Braun and Clarke's framework, ensuring credibility, dependability, and confirmability through independent reviews and consensus-based coding. Firstly, deep familiarization with the data was done by researchers. They thoroughly read and re-read the transcripts, listened to the recordings and initial notes were made. Codes were generated by using a deductive approach. color-coded and organized into potential themes. Sub-themes were formulated after extracting data from the narrative excerpts. A consensus by all researchers was reached on every step and discrepancies were resolved after discussion. Three main themes were identified: healthcare professionals are filling gaps beyond their roles or responsibilities, compelled to care despite several bottlenecks, and dissemination of awareness about nutrition care. Ambiguity in the role of healthcare providers in delivering nutrition care, a lack of in-depth knowledge of disease-specific nutrition care, and sub-optimal communication and counseling due to insufficient structural and economic resources may limit healthcare professionals' ability to impart nutrition care.

**Data availability statement:** All relevant data are within the paper and its Supporting Information files.

**Funding:** The authors received no specific funding for this work.

There is a critical need to enhance nutrition care delivery through improved nutrition education, strengthened interdisciplinary collaboration and referral systems, effective stakeholder engagement, and the implementation of system-level interventions. Addressing these obstacles can improve both the quality and accessibility of nutrition care, leading to better health outcomes.

## Introduction

Nutrition care refers to a comprehensive approach that includes the provision of nutrient delivery, dietary education, and meal services to prevent or manage nutrition-related conditions in clinical and public health settings [1].

Proper nutritional care supports healthy growth and development in all life stages while reducing the risk of malnutrition and non-communicable diseases (NCD). The consistent implementation of nutrition care is a multifaceted process requiring effective collaboration and teamwork among healthcare providers [2].

The scope of nutrition care encompasses various activities and interventions, ranging from individual-level care to population-wide initiatives. The nature of these activities often varies depending on the setting and the role of the provider. For instance, in hospital and intensive care settings, nutrition care may include medical nutrition therapy (MNT), delivered via oral intake, enteral nutrition, parenteral nutrition, or a combination thereof, depending on the patient's condition, dietary history, age, and level of malnutrition [3]. On the other hand, in community and long-term care facilities, nutrition care focuses on nutritional screening, education, assessment, medical nutrition therapy, monitoring and evaluation, supplemental intake, and proper documentation. Nonetheless, improved dietary behavior remains a cornerstone of nutrition care, contributing significantly to population health systems [4].

Nutritional well-being is supported not only by direct care providers like nutritionists and dietitians, experts working in clinical, food service, and public health settings who deliver evidence-based, systematic nutrition care [5]. Additionally, supportive care providers who influence the food environment are also included. The food environment, which consists of the physical, economic, political, and sociocultural factors that affect the availability, accessibility, and promotion of foods, plays a crucial role in shaping dietary choices and is an essential complement to nutritional care [6].

Although dietitians are primarily responsible for promoting effective and high-quality nutrition care, other healthcare professionals, such as doctors and nurses, play a significant supplementary role in delivering it. As the first point of contact in primary care settings, they are often in a position to recognize early signs and symptoms of nutrition-related health issues. This proximity places an ethical duty on doctors to provide sound nutritional guidance to support better health outcomes. However, studies have shown that many physicians lack the necessary knowledge and skills to confidently offer nutrition care, limiting their ability to adequately meet patients' nutritional needs. [7,8].

Correspondingly, midwives and nurses participate in nutrition care, but studies show that many lack the knowledge and confidence required to provide optimal

nutritional support [9,10]. In the hospital setting, nurses can play a pivotal role in malnutrition prevention by conducting nutritional screenings, monitoring dietary intake, addressing barriers to nutrition care, and ensuring the timely delivery of prescribed interventions. They are also responsible for assisting patients with meal consumption, arranging food delivery during off-hours, and supporting those unable to feed themselves [11]. However, evidence suggests that poor knowledge, inadequate role clarity, communication gaps, and insufficient meal assistance are often barriers to effective nutrition care in hospital settings, for which collaborative efforts and shared understanding among healthcare professionals are required to overcome these challenges [12].

Globally, inadequate nutrition care contributes significantly to mortality and disability associated with NCDs, with projections suggesting more than five million deaths and about 176.9 million disability adjusted life years (DALYs) due to NCD-related malnutrition by 2025 [13]. Despite this fact, the importance of nutrition care is often undermined, and the limited hiring of dietitians worldwide compromises the quality and comprehensiveness of patient-centered care. Studies from developed countries reported that despite knowing the fact of the adverse health outcomes of inadequate nutrition care, hospitals, patients, health professionals, and policymakers at large remain complacent in incorporating comprehensive nutrition care in healthcare settings [14,15].

In South Asia, nutritional care remains inadequate, with a high prevalence of malnutrition, especially among young girls, mothers, and children, emphasizing the need for urgent action, since early preventive measures against malnutrition may significantly reduce the burden of NCD in this region. However, literature suggests that clinicians rarely discuss dietary interventions with their patients, with insufficient nutrition knowledge being one of the major factors preventing clinicians' engagement in meaningful nutrition care provision [16].

Nutrition plays a central role in the prevention and management of noncommunicable diseases (NCDs). Poor dietary habits, such as high salt intake, contribute to hypertension, while excessive consumption of added sugars and unhealthy fats is linked to obesity, type 2 diabetes, and heart disease [17]. Pakistan, one of the world's most populous countries, faces a double burden of malnutrition and NCD, which account for over 50 percent of deaths in the country. On one side, undernutrition and micronutrient deficiencies remain common, while on the other, cases of overweight, obesity, and diet-related NCDs are rising rapidly [18,19]. Due to the multifactorial involvement in the current nutritional health situation of Pakistan, a lack of knowledge or misconceptions regarding nutrition care remains a major barrier to improving nutrition care outcomes, and instead of ineffective interventions targeting isolated factors, it is essential to understand the complex interplay of nutrition care determinants broadly [20,21]. The comprehensive understanding of the current nutritional status in Pakistan requires in-depth nuance to explore complexities in the perceptions and experiences of those involved in the provision of nutrition care in Pakistan, ultimately improving nutrition care policies and practices. There is a lack of published data in Pakistan regarding the comprehension and experiences of healthcare professionals regarding nutrition care provision. Findings from this study will provide an evidence base for potential structured and culturally appropriate nutrition education for healthcare professionals in Pakistan, with the potential to transform Pakistan's nutritional landscape in the future, ensuring better health outcomes and a more robust healthcare system.

## Materials and methods

This study utilized an exploratory qualitative research design to investigate healthcare professionals' understanding and experiences related to nutrition care. Ethical approval for this study was obtained from the Ethical Review Committee (ERC) of Ziauddin University, with reference code 7580823NTNUT. All authors hold advanced degrees in public health or related fields, including human nutrition and dietetics. At the time of the study, the authors served as university researchers and/or public health academics, with one researcher who had extensive experience in qualitative studies. The interviewers and analysts were all female. Data was collected through in-depth interviews with healthcare providers to ensure comprehensive coverage of the subject matter. Participants included fifteen adult doctors and nurses aged 18 years and older who provided indirect nutritional care to patients. However, dietitians and nutritionists, who are essential nutrition

care providers, were excluded. A purposive, convenient sampling technique guided participant selection to ensure the inclusion of relevant individuals working in various private and government hospital settings.

### Recruitment

The recruitment process involved disseminating study information through an on-site hospital notification system. The researchers contacted interested individuals via phone, their details were recorded, and they were scheduled for the interview. Participants were informed about the study's aims, voluntary participation, and the researchers' affiliation. They were provided with written informed consent, and before starting the interview, the informed consent was read aloud in the local language by the interviewer. Recruitment began on 19th December 2023 and was completed by 19th February 2024, while interviews ran concurrently and continued until April 2024. Signs of data saturation were noted during the later stages of data collection as recurring themes emerged, and remaining scheduled interviews were completed to confirm saturation and ensure data richness.

### Data- collection

Data collection engaged a semi-structured, open-ended guide that was developed, reviewed by an expert, and pilot tested by two trained researchers (S3 File).

The primary focus areas of the interviews included participants' understanding of nutrition care, their experiences in providing nutrition care to individuals or groups, issues and challenges encountered in providing nutrition care, and opinions of healthcare providers in improving quality and access to nutrition care in the Pakistani context.

The interviews were carried out from February 2024 to April 2024 in Karachi, Pakistan. Around three-fourths (n = 11) of the interviews were conducted at the participant's workplace in local language by researchers. No one else was present except the participant during the interview process. Four interviews were conducted over phone calls [5]. After conducting a total of n = 15 interviews, saturation of the data was reached when no new themes emerged upon further analysis.

Interviews typically lasted around forty minutes and were conducted in a conversational manner, either in person or over the phone. No repeated interviews took place, and interviewers used effective probing techniques to clarify vague concepts and gain deeper insights into participants' perspectives. Data collection continued until no new themes emerged, indicating saturation. We adhered to Lincoln and Guba's criteria [22] for 'trustworthiness' by establishing credibility through open-ended questions, dedicating ample time to the data, and providing a comprehensive description of our methods. Transcripts were not returned to participants, but the key findings were shared with selected individuals for validation. Audio recordings of the interviews, with participant consent, ensured accurate data collection. Audio recording was done by 2 different audio recorders after obtaining consent from participants and data collection accuracy was ensured. Audio files were password- protected, anonymized by removing participant names and place of work with registration number and access was only restricted to the research team members. Field notes were taken to capture nonverbal cues of the participants during an interview which helped in analysis and interpretation of data. Verbatim transcripts were anonymized and translated into English for comprehensive analysis.

### Data analysis

The researchers analyzed by becoming deeply familiar with the dataset by reading, re-reading the transcripts, documents and notes, listening to audio recordings multiple times to capture context, tone and nonverbal cues, then transcribing the text line by line for coding. Verbatim was transcribed, participant names and place of work were anonymized by registration number and in order to maintain consistency transcription protocols were used with set rules so all interviews were transcribed in the same way likewise the data was color coded by a deductive approach. The codes were then categorized into subthemes and finally into main themes. No software was used; themes were identified directly from the data.

To enhance trustworthiness and address validity concerns, we employed a thematic coding approach inspired by Braun and Clarke's work [23]. Starting from data familiarization, then assigning initial codes and themes generation after which the themes were reviewed, defined, named and lastly final production of the report after analysis was generated. (S4 File). To ensure transferability and contextual relevance, we included detailed descriptive data and direct quotes from participants within the text. For dependability, two of the authors independently reviewed, verified, and coded each line of transcription, with final interpretations achieved through consensus among all authors. To support confirmability, we included rich participant quotes that illustrate each theme. Consolidated criteria for reporting qualitative studies (COREQ) were pursued (S2 File) [24].

## Results

### Participants

Out of 15 participating healthcare professionals, eight were nurses while seven were doctors, including 11 females and four males who participated in the study, representing both genders. Participants ranged in age from 28 to 59, with 3–20 years of experience employed in Karachi's varied private and government hospitals.

### Emerging themes and sub-themes

Data analysis identified three main themes and nine sub-themes that demonstrated how healthcare professionals perceive nutrition care and their experiences delivering it in healthcare settings. The first theme was going beyond their responsibilities to fill gaps. The second theme involved a strong drive to care despite various bottlenecks, including role differences between public and private healthcare, communication barriers, limited knowledge about nutrition care, financial challenges, and lack of networking and collaboration. The third theme focused on disseminating awareness about nutrition care, including integrating nutrition education into curricula, establishing multidisciplinary collaboration and referral systems, spreading information, and engaging stakeholders. To enhance a deeper understanding of the key themes, relevant excerpts from the interview transcripts have been included.

### Theme 1: Filling gaps beyond their responsibilities

There was a consensus, with health care professionals acknowledging the role of nutritional care in patient management and recovery. They perceived nutrition counselling as an important aspect for patients' care and affirmed that nutrition plays a vital role in disease prevention and treatment, particularly in conditions such as diabetes, cardiovascular diseases, gastrointestinal disorders, and anemia. One of the female nurses emphasized the role of nutrition in patient management and stated:

"*Discussing diet is essential because many diseases are directly linked to nutrition. Today, conditions such as calcium deficiency and anemia are increasingly common. In the past, anemia cases were rare, but now, while blood transfusions may not always be necessary, iron deficiency is frequently observed, particularly among women and children.*" (Female Nurse-001)

Physicians were also convinced that nutrition is an integral component of patient management, and medication alone may not be the most effective way of improving patients' health outcomes. One of the male doctors shared his thoughts as follows:

"*Nutrition care is particularly important. Over the past two years, I have noticed that modifying a patient's diet according to their condition yields substantially better results, which would not be possible with medication alone.*" (Male Doctor-007)

A majority of the doctors reported that they are responsible for tracking weight changes, dietary habits, and patient counseling, while nurses stated that they monitor food intake and provide patient education.

One of the doctors emphasized the physical attributes of a patient as a proxy to assess his/her nutrition status:

"*During follow-up visits, typically after 2 weeks, we assess the patient's weight to gauge their progress. Weight gain provides valuable insight, along with the patient's overall physical appearance, helping us evaluate their nutritional status and overall well-being.*" (Female Doctor-001)

Some of the health professionals affirmed that they were directly responsible for providing general dietary advice for patient education and helping patients understand the basics of healthy eating habits, irrespective of their (health professionals) level of knowledge and confidence to provide nutrition care. A few of the physicians affirmed that they were shouldering the responsibility of providing nutrition counselling in the absence of dietitians in healthcare settings, while some of the participants considered that the nutrition care provision role should be exclusively confined to dietitians or nutritionists. Therefore, with regards to the perception of participants in the provision of nutrition care and counselling, the majority of the respondents opined that the responsibility for the provision of nutrition care in healthcare settings should ideally be on trained nutritionists or dietitians. According to one of the male nurses:

"*Dietitians are the best guides when it comes to nutrition care, as they have comprehensive knowledge, structured meal plans, and a detailed insight into dietary needs. Their expertise is crucial in helping patients to follow an appropriate diet for better health outcomes. Since dietitians have studied nutrition in depth, they play a principal role in ensuring patients adhere to proper dietary guidelines. In contrast, doctors chiefly focus on medical treatment and prescriptions, making dietitians the most qualified experts to provide specialized nutrition guidance.*" (Male Nurse-006)

Notably, it was also recognized by the participants that due to some technical constraints, the provision of nutrition care and counseling by nutritionists and dieticians is not always possible. Besides, a few of the respondents also opined that since the patients trust physicians and nurses, nutrition counselling should be provided by these healthcare providers. One of the female doctors verbalized her opinion and said:

"*… if patients are on warfarin, they require proper counseling. However, dietitians and nutrition counselors are not always readily available. So, it becomes the responsibility of the doctor to provide effective advice. A truly skilled doctor should also be a good counselor.*" (Female Doctor-001)

The same healthcare provider further emphasized the importance of involvement of physicians in providing nutrition care and counseling and stated:

"*A healthcare professional should acquire both profound academic qualifications to develop a deep understanding of nutrition and empathy to connect with patients and address their concerns effectively, since they are trusted by their patients*" (Female Doctor-001)

### Theme 2: Compelled to care despite several bottlenecks

**Role variability between public and private healthcare settings.** There was variability in the extent of involvement of different health care professionals in the assessment, monitoring, and counseling of patients while providing nutrition care. There were also disagreements in nutrition care and counseling provision between public and private healthcare settings. In private hospitals, dietitians work closely with doctors, which ensures inclusive dietary supervision and care. However, a lack of structured referral systems in public healthcare settings often hinders interdisciplinary teamwork,

impeding patients from accessing specialized nutrition care. Additionally, the limited availability of dietitians in public healthcare settings results in health workers taking on the duties of nutrition counsellors, even though they may lack advanced expertise in dietetics.

In private healthcare settings, dietitians usually tend to work in collaboration with doctors and other staff members, as one of the female doctors working in a private healthcare facility stated:

"*In our facility (private setting), nutritionists are readily available, and in addition to their guidance, we also counsel patients ourselves. As a result, patients receive dietary counseling two to three times, reinforcing the importance of nutrition in their treatment. When patients hear the same advice multiple times, they are more likely to recognize its significance and follow it diligently.*" (Female Doctor-003)

Similarly, one of the female nurses working in a private healthcare setting also affirmed the involvement of dietitians in providing nutrition support to patients:

"*In terms of food and nutrition, any surgery patient, whether bedridden or suffering from anemia, requires proper dietary guidance. In such cases, we reinforce the doctor's recommendations, ensuring that patients follow the prescribed dietary guidelines for better recovery and overall health.*" (Female Nurse-001)

On the contrary, another female doctor working in a public healthcare setting stated:

"*Currently, there are no dedicated nutrition counselors available. As a result, when a patient needs dietary guidance, the responsibility falls entirely on me. It is, therefore, my duty to counsel all patients effectively and address their concerns individually.*" (Female Doctor-001)

However, a few health professionals solely relied on a dietitian's specialized medical nutrition therapy. One of the male nurses, working in a private healthcare setting, reported:

"*Dietitians in the hospital conduct daily morning rounds, assessing each patient's condition, and we also ensure that the prescribed diet aligns with the patient's specific nutritional requirements and method of intake.*" (Male Nurse-006)

**Communication barriers.** Participants agreed that time constraints, extreme patient loads, and heavy workloads often limit the extent of dietary advice, forcing doctors to prioritize medical treatments over comprehensive nutrition consultation. Additionally, language and communication hurdles further complicate effective counseling, particularly when patients have low literacy levels or face language difficulties in understanding dietary recommendations. One of the female nurses shared her experience of facing challenges in the provision of nutrition care:

"*Patients in this area came from an underdeveloped region, which presents several challenges. The most significant issue is the language barrier, making communication difficult. Additionally, many patients lack awareness and understanding of their health conditions. Despite our best efforts to educate and counsel them, they often insist on receiving medication, believing it to be the only solution for their ailments. This mindset makes it tough to emphasize the significance of diet in their treatment. Although we strive to shift their focus from solely relying on medicine to prioritizing proper nutrition, overcoming deeply ingrained beliefs requires considerable time and effort.*" (Female Nurse-004)

Despite the proven distinction of patient-centered nutrition counselling in improving informed decision-making about nutrition, it was evident that the decision-making about the provision and consumption of nutrition mainly lies with doctors or dietitians. One of the male nurses working in a public healthcare setting commented:

*"No, patients usually do not ask us about their dietary needs. Instead, doctors make these decisions."* (Male Nurse-006).

Another female doctor opined that even though patients are not actively engaged in decision-making about their own nutrition, healthcare providers should have empathy and compassion while making decisions on the patient's behalf:

*"It is important to maintain a compassionate approach, as medical professionals can sometimes come across as harsh. At the very least, healthcare providers should ensure that their interactions with patients are supportive and respectful"* (Female Doctor-001)

**Lack of in-depth knowledge regarding nutrition care.** Most of the participants identified a lack of knowledge and skilled training of healthcare professionals as a paramount obstacle in providing effective nutritional care. Participants also declared that medical and nursing programs of study ignore adequate coverage of nutrition, and that healthcare professionals often have limited knowledge and confidence in dietary counseling. They frequently struggle to provide evidence-based recommendations or to give nutrition advice according to patient needs. One of the female doctors working in a public healthcare setting commented:

*"We lack in-depth knowledge about food and nutrition, and as a doctor, I have neither practiced it extensively nor observed its implementation, especially in Pakistan. As a result, nutrition and diet-related knowledge remain quite superficial. But I do understand basic dietary recommendations…. Such as advising a diabetic patient that they can consume one fruit per day and should avoid bakery items."* (Female Doctor-002)

Another doctor acknowledged a lack of in-depth knowledge about the specific nutritional requirements of patients and stated:

*"As doctors, our learning in this area is quite limited for the reason that we are so deeply focused on medicine, we hardly prioritize nutrition. The available information resources are also scarce, what we know mostly comes from childhood advice passed down by our parents, such as 'don't eat this' or 'avoid that.' As a result, doctors usually have only a superficial understanding of nutrition and diet"* (Female Doctor-006)

**Financial constraints.** Most healthcare professionals felt that financial instability and limited access to nutrient-rich foods create obstacles to nutritional care. Several patients struggle to afford healthy diets or specialized dietitian consultations, particularly in low-income communities. One of the nurses addressed this issue and commented:

*"There are still people in 2024 who cannot afford essentials, with some surviving on just one meal every 48 hours. If employment opportunities were more stable, individuals would at least be able to provide for their families daily. Inflation remains a major hurdle, making fundamental food items increasingly expensive. As prices continue to rise, many households are unable to buy the necessary food items, while those who can afford them are forced to compromise—where they might need 1 kg of an item, they settle for much less"* (Male Nurse-008)

Moreover, patients often prioritize medication over dietary changes, making compliance with nutrition recommendations difficult. Healthcare providers acknowledge the significance of nutrition counselling for better health outcomes. However, they also reflected on the fact that financial constraints may prevent a patient from seeking professional nutrition counselling:

*"Everything comes at a cost, and in Karachi, Pakistan, our healthcare system is not insurance-based, nor is it backed by the government. As a doctor, I must also think about the financial burden on my patients. Referring them to multiple*

*specialists increases their expenses, making it difficult for those paying out of pocket to afford multiple consultations. For example, if a physician suspects a heart problem, they may refer the patient to a cardiologist, while also recommending them to consult a dietitian for proper nutrition guidance. However, for many patients, managing the cost of visiting multiple specialists can be a considerable challenge.”* (Male Doctor -006)

**Lack of health literacy.** Additionally, a lack of health literacy can make it difficult for patients to absorb and implement dietary suggestions, leading to poor adherence to nutrition plans. Cultural food preferences, misconceptions about nutrition, and resistance to lifestyle changes further contribute to non-adherence to dietary advice.

*“Guiding patients can at times be challenging because they often have preconceived beliefs about certain foods. For instance, if we recommend bananas, some patients express concerns that it may cause chest congestion. This ambiguity makes it difficult to determine whether bananas should be advised or avoided, even though there is no medical evidence supporting such claims. Similar fallacies exist about rice and other foods, which are commonly repeated but lack scientific justification. So, nutrition counseling requires significant time and effort to address these concerns and educate patients successfully.”* (Female Doctor-002)

*“When we first explain something to patients, they often struggle to accept it because they do not have the right mindset. Initially, they follow very little, if anything at all. However, as complications arise, their inclination to listen gradually increases. Over time, their trust in the advice grows, and they begin to understand the importance of following proper guidance.”* (Female Doctor-001)

**Lack of networking and collaboration.** Besides structural and financial barriers, a lack of collaboration and weak network linkages along with a lack of initiative on part of the dietitians and nutritionists to reach out to patients, reason being high patient load and few hiring of dietitian with less pay scale were also identified as potential barriers to improve patient-centered nutrition in Pakistan. One of the doctors working in a private facility commented:

*“There are dietitians available in the hospitals where I have worked. However, their participation often depends on referrals—if a patient is assigned to them, they take over the case, but otherwise, they do not actively reach out to patients on their own.”* (Male Doctor-006)

Participants agreed that a well-functioning nutrition care system requires clear role definitions and improved dietitian integration in clinical teams. Furthermore, strong communication skills, empathy, and professional qualifications are essential attributes for effective nutrition care providers, ensuring that patients receive accurate and practical dietary guidance.

*“I believe dietitians and community nurses can work effectively together in promoting nutrition awareness. Like, they can collaborate to organize community seminars and set up designated settings, just as we have done here, to educate people about healthy eating habits. Even if direct assistance is not possible, providing basic nutritional guidance can still have a meaningful impact on public awareness and overall well-being.”* (Male Doctor-008)

### Theme 3: Disseminating awareness about nutrition care

**Integrated curriculum for nutrition education and training.** Participants suggested enhancing the quality of nutrition care provision by integrating nutrition-focused training programs within medical and nursing curricula. One of the female doctors working in a private healthcare facility emphasized the significance of imparting nutrition education to medical doctors, as continuous professional development through workshops, seminars, and continuing medical education (CME) programs can equip them with the necessary skills and knowledge to offer accurate dietary guidance:

*"Doctors should also have nutrition guidance and education at their professional level. In any healthcare setup that includes a nutritionist, there should be monthly or bi-monthly meetings between doctors and nurses to enhance nutrition awareness. In interdisciplinary hospitals, it is essential to discuss nutrition as part of regular morning meetings, as dietary choices play a crucial role in overall health and significantly impact patient outcomes."* (Female Doctor-5)

One of the doctors working in a public healthcare facility stated on the importance of engaging influential stakeholders in strengthening nutrition care in Pakistan by utilizing available resources:

*"Our senior officers, such as the medical superintendent and district health officer (DHO), play a crucial role in advancing nutrition awareness. To ensure progress, we must first educate them on the magnitude of this issue. The DHO oversees all towns and medical superintendents, and once he realizes the significance of nutrition, he can efficiently mobilize work across the district. With access to resources and authority, he can set up health camps, distribute pamphlets, issue official circulars, and mandate nutrition awareness programs. Engaging with officials at the district level can streamline implementation, making it significantly easier to integrate nutrition programs into the healthcare system."* (Female Doctor-001)

**Establishing a multidisciplinary collaboration and referral system.** Participants reported that systematic, effective nutrition care requires coordinated efforts among all healthcare professionals. Establishing structured referral systems can ensure that doctors, nurses, and dietitians work together seamlessly to provide patients with thorough nutrition guidance. Regular case discussions, joint consultations, and interdisciplinary meetings can facilitate interaction among professionals and upgrading progression of care. Additionally, healthcare institutions should work towards increasing the availability of dietitians in hospital settings for professional nutritional care practice and nutrition education for systematic health programs.

*"Particularly for nutrition counseling, there should be a dedicated professional. If only a single counselor is available, we should conduct group sessions with three or four patients at a time, this will make the process more efficient and convenient for both the counselor and the patients. To enhance the effectiveness of nutritional education, organize structured sessions or involve multiple lady health workers. Additionally, even within a small town, a qualified dietitian should hold at least one or two sessions per week. These sessions can also include health workers, allowing them to assist in patient education while providing valuable awareness about established nutrition-related issues in different areas. Each town should have at least one dedicated dietitian or nutritionist who can supervise and conduct training sessions for health workers. This collaborative approach would ensure that health workers are well-equipped to support nutrition counseling while also keeping dietitians informed about common dietary concerns in the community."* (Female Doctor-001)

**Spreading information and stakeholders' engagement.** Participants advocated the use of technology and public health initiatives to enhance nutrition awareness and accessibility. Digital platforms like mobile apps, social media campaigns, and virtual counseling services can provide widespread nutrition education to both healthcare providers and patients.

*"I believe social media is the most effective platform for spreading nutrition awareness. It grants easy communication with a wide audience, making information accessible to people from all environments. Posting short, engaging messages about nutrition, such as those related to carbohydrates, potassium, or essential vitamins and minerals, can drastically enhance public knowledge. In addition, short educational videos can be an excellent mode to increase awareness, as they are easy to watch and understand. Unlike seminars, which often have limited turnout due to time*

*constraints, social media reaches a much larger audience. If nutrition content is presented in Urdu and simple, lay audience friendly language, it would be more effective in educating people across different levels of society.*" (Female Doctor-004)

Some health professionals insisted on community-based nutrition programs, school-based education, and local outreach strategies for addressing dietary knowledge gaps in underserved populations. Moreover, policy interventions encouraging insurance coverage of dietitian consultations and financial assistance programs for low-income patients can improve access to specialized nutrition care, ensuring that more individuals benefit from professional dietary guidance.

"*Various stakeholders, including hospital departments, state agencies, government and private sectors, and NGOs (non-government organizations), can play a crucial role in promoting nutrition awareness. NGOs, government bodies, and private institutions can make a significant impact. Furthermore, multinational companies can contribute through corporate social responsibility initiatives. Community outreach efforts, such as door-to-door distribution of pamphlets with brief five-to-ten-minute awareness discussions, can be effective. Placing informational pamphlets in high-traffic areas like markets, restaurants, and fast-food outlets can further enhance visibility. For children, broadcasting educational content through cartoons and social media platforms like WhatsApp can make learning about nutrition engaging and accessible. Executing these strategies can lead to greater public awareness and improved health outcomes.*" (Male Doctor-007)

## Discussion

Evidence-based nutritional care in healthcare setups has been under-researched [25,26] This study was expected to explore the perception of health care experts, nurses, and doctors working in various hospitals regarding nutritional care. It also investigated the perceived role, barriers, and facilitators in providing nutritional care. The study identified three main themes: filling gaps beyond their responsibilities, compelled to care despite several bottlenecks and disseminating awareness about nutritional care from the perspective of healthcare professionals.

Recent findings from our study acknowledged the importance of nutrition and the challenges healthcare professionals face in delivering effective nutrition care. Professionals generally recognize that nutrition plays a crucial role in patient recovery and management. They are aware that numerous health conditions are influenced by dietary factors, a lack of proper delivery causes severe health complications, and restricts optimal health. This understanding aligns with the World Health Organization's (WHO) assertion that poor nutrition significantly contributes to non-communicable diseases (NCDs), which account for approximately 71% of global deaths [2].

Healthcare providers are increasingly perceived to be responsible for educating patients on the importance of a balanced diet and early intervention strategies to mitigate risks associated with diet-related conditions. Despite participants feeling a lack of confidence, competence, or unpreparedness, they are obligated to provide nutrition care. Subsequently, many healthcare professionals expressed concerns regarding their limited training and knowledge in nutrition, which can impede their ability to offer effective dietary advice. There is a clear need for enhanced nutrition education within medical and nursing curricula; integrating comprehensive nutrition training into these programs would better equip future healthcare providers with the necessary skills to address nutritional concerns confidently [27,28].

This study has highlighted perspectives of health experts on their roles in delivering nutrition care and the critical gaps in understanding nutrition and evidence-based practice [29]. Participants primarily contribute to nutrition care delivery through assessment, monitoring, and counseling patients on common issues such as diabetes, skin diseases, and nutrient deficiencies. Conversely, assessments were mainly conducted through clinical observation, and weight changes were used as a proxy; evidence-based assessment methods were rarely incorporated into the nutrition care process by them.

The guiding strategies employed by participants predominantly focus on general healthy eating advice but do not consistently align with established nutrition-related recommendations or professional nutrition care.

Dietitians were typically found in private hospitals, often absent from government healthcare settings. Despite the limited availability of dietitians, healthcare professionals feel compelled to provide nutrition counseling and served as the sole decision-makers, operating without a structured framework for nutrition care, even though their expertise or role clarity in this area was limited, which was identified from our study. From the evidence, where health experts of multidisciplinary full-time nutrition support teams (NST) (including dietitians, physicians, nurses, pharmacists) in ICU settings for patients with acute respiratory distress syndrome (ARDS) were associated with a 19% lesser a month mortality and 12% lower annual mortality compared to ICUs without NST involvement [30]. Moreover, implementing NST for in-hospital parenteral nutrition has been found to reduce catheter-related infection rates, less inappropriate parenteral nutrition use, and modest reductions in mortality [31]. These comparisons emphasize the necessity for a multi-layered model in our health system that establishes decision making protocols, streamlines processes, clarifies role differentiation, and promotes collaboration among healthcare professionals for a more inclusive approach to improve nutrition care and patient outcomes.

Participants recognized professional knowledge, practical experience, and communication skills as essential characteristics for dietetic practice. Dietitians are considered the most appropriate individuals to deliver nutritional care. Empathy and strong communication skills are particularly important as they foster a better understanding of patients' individual needs, establish trust, and ensure effective patient-provider interactions. Training programs should emphasize these soft skills alongside technical knowledge to produce well-rounded nutrition care professionals. According to the Academy of Nutrition and Dietetics (AND), dietitians and nutritionists are shaped by their foundational education and supervised practice, which is further enhanced through ongoing learning opportunities such as advanced degrees, continuing education, and specialized training certificates [32].

Conversely, standards within the nutrition care profession remain underdeveloped in many developing countries. The accreditation system, ensuring the quality of academic and professional training in this field, is lacking. A review of the history and development of dietitian recognition systems emphasizes the qualifications and practices required for experts in nutrition and dietetics [33].

Numerous bottlenecks impede the effective provision of nutrition care. It may be due to systemic issues such as inadequate staffing and a shortage of dietitians in healthcare settings, time constraints, and heavy workloads that limit the healthcare professional's capacity to engage meaningfully with patients about nutrition. Additionally, socioeconomic factors may restrict patients' access to nutritious foods, complicating adherence to dietary recommendations. Also, language barriers can hinder effective communication between providers and patients, highlighting the need for culturally competent care that accommodates diverse patient backgrounds.

In this study due to various food-related myths and behavioral issues regarding dietary changes, health workers mostly find it hard to deliver effective counseling. This findings align with a systematic review conducted on patient's with type 2 diabetes nutrition care provision in Africa identified the systemic and individualized barriers such as lack of knowledge, poverty, high costs of diabetes care, and limited access to healthcare as significant obstacles in preventing and managing diabetes [34].

Although, to enhance nutrition care provision, healthcare professionals have proposed several strategies for improving counseling and education efforts for both providers and patients is paramount, utilizing technology to facilitate access to nutritional information can empower patients, fostering community engagement through partnerships with local organizations can enhance resources available for nutritional support; encouraging shared decision-making between patients and their healthcare teams can promote greater adherence to dietary recommendations [35,36].

Though dietitians/nutritionists exist to some extent within private healthcare systems, general practitioners are typically the first point of contact for patients. In many healthcare settings, referrals from doctors are necessary for consultations with nutritionists; therefore, referrals depend on doctor's understanding of nutrition care. However, inconsistent levels of

nutritional knowledge among health experts can lead to limited referrals and coordination gaps between health professionals and dietitians/nutritionists [5].

This study identified several fundamental challenges like those explored previously; these hindrances can only be addressed through policies focused on nutritional care. To alleviate the burden of NCDs on the healthcare system, behavioral change can be fostered by enhancing health experts' confidence and self-efficacy through improved knowledge [8].

Qualitative research conducted among Australian and UK-based doctors indicates a demand for a shift from medical treatment towards preventive care to enhance coherent application of comprehensive nutrition care. This shift can be achieved by improving health experts' perceptions of nutrition care through enhanced educational models [37]. Another study conducted in Pakistan demonstrated that inadequate nutrition care by healthcare providers may stem from limited knowledge and the absence of clear guidelines. Giving specialized nutrition training and ongoing professional development (CPD) can significantly enhance their knowledge, attitudes, and practices (KAP) related to nutrition [38].

While training healthcare practitioners in basic nutrition care is important, it is equally necessary to recognize that many are already overburdened and face significant time constraints in providing adequate nutritional support. Evidence suggests that training alone is insufficient to improve nutrition outcomes unless supported by structural reforms [39,40]. A system-level change is essential, one that integrates nutritionists and dietitians more fully into patient care within health services. Training should emphasize when and how to refer patients to dietitians, ensuring that nutrition specialists are consistently available as part of an interdisciplinary team. Such an approach has been evident to improve patient outcomes, reduce complications, and ease workload pressures on non-specialists, developing a more effective, patient-centered model of nutrition care [41,42].

Therefore, addressing these bottlenecks through targeted education provides a clear understanding of nutrition care, interdisciplinary collaboration, and system-level integration of dietitians as core members of healthcare teams within healthcare settings. This will not only provide a sustainable, interdisciplinary approach but also tackle the challenges of malnutrition and NCDs within the population.

## Strengths and limitations

To the extent of our knowledge, this novel study was the first to qualitatively explore the concept of nutrition care among healthcare professionals. A notable strength of this study is its consideration of the perspectives of health professionals operating across various levels of medical care. Although the moderately small sample size (n = 15) may be viewed as a limitation, it fulfilled the criteria for a qualitative study and achieved theoretical saturation. Additionally, the study's focus on participants from Karachi, Pakistan, necessitates caution in generalizing the findings, as differences in healthcare systems and population sizes could influence the insights gained. The involvement of two dietitians among the researchers may have influenced data interpretation to emphasize nutrition care as a solution in healthcare. However, one researcher who is not a dietitian or nutritionist contributed to an independent analysis, and themes were finalized after the discussion and consensus of all the authors of this study. While a few phone interviews may have restricted the observation of non-verbal cues, this format allowed for the recruitment of experts from diverse locations [30]. Future research could explore the perspectives of nutritionists and dietitians to systematically examine gaps in patient engagement and strengthen the provision of nutritional care. In addition, it would be valuable to explore other stakeholders of nutrition care and identify opportunities for strengthening the delivery of health care.Click or tap here to enter text..

## Conclusion

This study emphasizes the perception about nutrition care in patient care and the barriers and facilitators among healthcare professionals. Addressing these challenges through enhanced education, interdisciplinary collaboration, and systemic changes is crucial for improving nutrition care quality. A deeper understanding of nutrition among healthcare

providers not only enhances patient outcomes but also aids in the prevention and management of chronic diseases across populations.

## Supporting information

**S1 File. Coding tree: Highlights the main themes, subthemes and their description.**
(DOCX)

**S2 File. ISSM COREQ (Consolidated criteria for Reporting Qualitative research) checklist.**
(DOCX)

**S3 File. Semi structured interview guide questions for healthcare professionals.**
(DOCX)

**S4 File. Thematic analysis framework.**
(DOCX)

**S5 File. Data file.**
(ZIP)

**S1 Checklist. Inclusivity in global research.**
(DOCX)

## Author contributions

**Conceptualization:** Naeema Tahira Syed, Sukaina Shabbir.

**Data curation:** Naeema Tahira Syed, Sukaina Shabbir.

**Formal analysis:** Naeema Tahira Syed, Sukaina Shabbir.

**Investigation:** Naeema Tahira Syed, Sukaina Shabbir.

**Methodology:** Naeema Tahira Syed.

**Supervision:** Maria Atif.

**Writing – original draft:** Naeema Tahira Syed.

**Writing – review & editing:** Naeema Tahira Syed, Sukaina Shabbir, Maria Atif, Rubina Hakeem.

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
