## [Decision Letter · Decision Letter 0]

10 Sep 2025

PGPH-D-25-01594

Qualitative exploration of comprehension and experiences of healthcare professionals regarding nutrition care in Karachi, Pakistan

Dear Dr. Atif,

Thank you for submitting your manuscript to PLOS Global Public Health. After careful consideration, we feel that it has merit but does not fully meet PLOS Global Public Health’s publication criteria as it currently stands. Therefore, we invite you to submit a revised version of the manuscript that addresses the points raised during the review process.

You will note that the reviewers were unanimous in their praise for your work and the quality of your paper. I'd ask you to please pay attention to the comments about clarity in the methods and to address Reviewer 2's comments thoroughly. 

We look forward to receiving your revised manuscript.

Kind regards,

Nicola Hawley

Academic Editor

Journal Requirements:

1. Please amend your online Financial Disclosure statement. If you did not receive any funding for this study, please simply state: “The authors received no specific funding for this work.”

2. Please update your online Competing Interests statement. If you have no competing interests to declare, please state: “The authors have declared that no competing interests exist.”

3. In this instance it seems there may be acceptable restrictions in place that prevent the public sharing of your minimal data. However, in line with our goal of ensuring long-term data availability to all interested researchers, PLOS’ Data Policy states that authors cannot be the sole named individuals responsible for ensuring data access (http://journals.plos.org/globalpublichealth/s/data-availability#loc-acceptable-data-sharing-methods).

4. “S2 Checklist.docx” is currently uploaded as an 'Other' file type, which is not viewable by reviewers. Please ensure that all files meant for review are uploaded as 'Supporting Information' and include a legend in the manuscript.

5. We have noticed that you have uploaded Supporting Information files, but you have not included a list of legends. Please add a full list of legends for your Supporting Information files before or after the references list.

Additional Editor Comments (if provided):

Reviewer #1:

Reviewer #2:

Reviewer #3:

Reviewer #4:

Reviewers' comments:

Reviewer's Responses to Questions

**Comments to the Author**

1. Does this manuscript meet PLOS Global Public Health’s publication criteria?

Reviewer #1: Yes

Reviewer #2: Yes

Reviewer #3: Yes

Reviewer #4: Yes

2. Has the statistical analysis been performed appropriately and rigorously?

Reviewer #1: N/A

Reviewer #2: N/A

Reviewer #3: N/A

Reviewer #4: N/A

3. Have the authors made all data underlying the findings in their manuscript fully available (please refer to the Data Availability Statement at the start of the manuscript PDF file)?

Reviewer #1: Yes

Reviewer #2: Yes

Reviewer #3: No

Reviewer #4: Yes

4. Is the manuscript presented in an intelligible fashion and written in standard English?

Reviewer #1: Yes

Reviewer #2: Yes

Reviewer #3: Yes

Reviewer #4: Yes

Reviewer #1: Comments

-The authors were silent on field notes, can they provide insights on how field notes were handled?

-During coding, how did authors resolve conflicts.

-What type of qualitative coding did the authors perform on their transcripts, kindly be specific on which was used as it may also influence your results and interpretation.

-As part of their report, authors should indicate how their backgrounds and experiences influenced the interviews, their codes, their themes, the results of this study, and how these were handled.

-The authors should kindly report about their personal biases, assumptions, reasons and interests in the research topic.

-Authors indicated that “Data was first immersed by the NTS and SS”, but it is unclear how many people did the coding. Authors should also be friendly with non-technical readers by using terms that can generally be understood.

-Authors should also consider structuring their work according to PLOS Global Public Health reporting guidelines.

-Additional comments are embedded in the word file.

Reviewer #2: Thank you for inviting me to review this manuscript. The topic is both timely and important, given the global burden of malnutrition. This research highlights a critical point and can further the discussion on the need for interdisciplinary collaboration in healthcare settings. In particular, the study’s main contribution is to bring forward healthcare practitioners’ perspectives on barriers to nutrition care. I offer the following constructive comments and suggestions to strengthen the manuscript:

- In the first sentence (lines 47–49), please provide a citation for the definition used. Is this sourced from a nutrition or dietetics manual, glossary, institutional document, or report?

- Page 4, line 65: When introducing the term ‘food environment,’ consider providing a brief definition and a supporting reference or example. This will clarify your meaning for readers, given that the introduction has focused on ‘nutrition’ until this point. Highlighting the importance of the food environment in the nutrition context would strengthen this section.

- Page 4, lines 77–78: Please add a reference for the ‘study’ mentioned regarding midwives’ and nurses’ participation in nutrition care and their reported lack of knowledge and confidence. Also, consider clarifying whether you are referring to a specific study, multiple studies, or research in general, for precision.

- In the introduction, you discuss the connection between NCDs and nutrition. It would be helpful to provide examples of the types of NCDs to which you are referring. Since ‘NCD’ is mentioned throughout the manuscript without specific examples, consider clarifying which NCDs are most relevant to your discussion—especially when describing the double burden of malnutrition and NCDs (e.g., in lines 101–102 regarding Pakistan). This will help readers better understand the relationships between nutrition and specific diseases.

- In the methods section (lines 150–155), the way participant numbers are presented is unclear and could be confused with citation style. Consider using ‘(n=11)’ and ‘(n=5)’ for clarity.

- The use of authors’ acronyms (e.g., NTS and SS) is confusing, especially in the data analysis section. I recommend removing these acronyms and instead using a general description of the process (e.g., ‘We analysed data by…’). Stating which author performed each task may not be necessary and could be omitted for clarity.

- Page 9, line 169: When referencing Braun and Clarke’s work, please include the appropriate citation for their thematic analysis method. The current citation appears to be for COREQ, which relates to reporting guidelines, not the analysis method itself.

- Results, page 13, lines 271–279: This section appears to address practitioners going beyond their responsibilities, rather than fitting within the private/public sub-theme. Providing additional context on why this content belongs in the sub-theme would be helpful.

- Page 21, line 447: If appropriate for the local context and translation, consider replacing ‘man’ with a more inclusive term such as ‘lay-audience.’ Please disregard this suggestion if it does not suit the language or cultural context.

- Discussion, first sentence: If available, please cite a systematic review or other source to support the claim that this area is under-researched, as it forms a key basis for your study.

- Lines 470–473: Currently, this section reads as though there are more than three themes. Ensure that the three themes match those described in the results (e.g., Beyond responsibility, Being compelled, and Disseminating).

- Lines 474–475: Please clarify whether the ‘recent findings’ refer to your own study or to the broader literature. If referring to external research, include a citation.

- Lines 499–507: To contextualize this important result, consider comparing it with a health system where this recommendation is implemented, or another setting where multidisciplinary approaches have been successful. Relating your findings to the broader literature would strengthen the discussion.

- In your discussion, you highlight the importance of training. However, you also note that health practitioners are often overloaded and face constraints in providing nutritional support. It may be worth suggesting a systems-level change that involves integrating nutritionists more fully into patient care, rather than simply increasing training for existing practitioners. Training could focus on when and how to refer to nutritionists, ensuring these specialists are available within public health services to support a truly interdisciplinary approach.

- In your ‘future research’ section, you note that some practitioners highlight gaps in the interaction between nutritionists/dietitians and patients. It may be valuable to explore the perspectives of nutrition professionals in future studies. Including this as a research need could help identify where gaps truly exist.

Minor edits:

- p.4, line 76: remove the period before references (5)(6).

- p.7, line 143: use “(Table 1)” for clarity.

- p.15, line 322, and p.16, line 348: capitalize “I.”

- p.21, line 552: “non-communicable disease (NCDs)” is already defined earlier; no need to repeat in full.

Reviewer #3: This work addresses a gap in the literature on the challenges faced in developing countries, and in Pakistan in particular, in the delivery of nutritional care. The results presented are consistent with the announced title and the methodology presented. The manuscript is written in a readable and understandable manner. However, some clarifications could be made regarding the methodology. It is not clear whether the recruitment and the interviews were carried out simultaneously or in a staggered manner as suggested by the dates mentioned because of this sentence "The recruitment process started on 19th December 2023. Shortly after, participant interviews began and continued alongside recruitment. Recruitment ended on 19th February 2024, once initial signs of data saturation appeared". How did the data saturation appear before having collected data?

Reviewer #4: The paper is well written. Additional nformation related to we paradigm that guided the study will and to the richness of the paper. Authorsshould also review punctuations around atactions esmee 75 seems to have a full stop before and after The citationsee unes 77 to 82 _ there's red to cold citations You have indicated in line 77 ' study show is it just one..cite which study or studies the statements that follow are poorly cited.

**Do you want your identity to be public for this peer review?** For information about this choice, including consent withdrawal, please see our Privacy Policy

Reviewer #1: No

Reviewer #2: **Yes: ** Marina Banuet-Martínez

Reviewer #3: No

Reviewer #4: **Yes: ** Ajike SO

---

## [Decision Letter · Decision Letter 1]

3 Nov 2025

Qualitative exploration of comprehension and experiences of healthcare professionals regarding nutrition care in Karachi, Pakistan

PGPH-D-25-01594R1

Dear Dr Atif,

We are pleased to inform you that your manuscript 'Qualitative exploration of comprehension and experiences of healthcare professionals regarding nutrition care in Karachi, Pakistan' has been provisionally accepted for publication in PLOS Global Public Health.

Best regards,

Nicola Hawley

Academic Editor

Reviewer Comments (if any, and for reference):

Reviewer's Responses to Questions

**Comments to the Author**

Reviewer #1: All comments have been addressed

Reviewer #2: All comments have been addressed

Reviewer #3: All comments have been addressed

Reviewer #4: All comments have been addressed

publication criteria?

Reviewer #1: Yes

Reviewer #2: Yes

Reviewer #3: Yes

Reviewer #4: Yes

3. Has the statistical analysis been performed appropriately and rigorously?

Reviewer #1: N/A

Reviewer #2: N/A

Reviewer #3: N/A

Reviewer #4: N/A

4. Have the authors made all data underlying the findings in their manuscript fully available (please refer to the Data Availability Statement at the start of the manuscript PDF file)?

Reviewer #1: Yes

Reviewer #2: Yes

Reviewer #3: Yes

Reviewer #4: Yes

5. Is the manuscript presented in an intelligible fashion and written in standard English?

Reviewer #1: Yes

Reviewer #2: Yes

Reviewer #3: Yes

Reviewer #4: Yes

Reviewer #1: I have no additional comments.

Reviewer #2: The edits have improved the manuscript, and I don't have further comments. There are only a few minor suggestions, some of which may be due to the PDF version I received:

- Please merge the first two sentences of the introduction into one paragraph.

- On line 83 and line 192, remove the final punctuation before the citations. Additionally, ensure that the citation style is consistent; the newly added references are formatted as (7)(8) instead of (7, 8). Please review this throughout the manuscript. If the journal corrects this during the editing process, feel free to disregard my comment.

- On line 108, there is no need to spell out "NCDs" again since it has already been defined earlier. Please review this throughout the manuscript.

- On line 162, my previous comment regarding the use of 'n=##' was intended for numbers in parentheses, given that your citation style is similar (e.g., (11) and (5)). However, since this line states "...a total of 15 interviews...", it is not necessary to include 'n=15'. This doesn't need to be edited; it is acceptable as it stands, but I wanted to offer this as a suggestion.(some might be because of the PDF version I received):

Reviewer #3: The discomfort expressed regarding methodological details during the initial evaluation has been clarified and corrected through reformulation. I am satisfied with the correction.

Reviewer #4: the paper has been strengthened and can be published as is

**Do you want your identity to be public for this peer review?** For information about this choice, including consent withdrawal, please see our Privacy Policy

Reviewer #1: No

Reviewer #2: **Yes: ** Marina Banuet Martínez

Reviewer #3: No

Reviewer #4: **Yes: ** Ajike Saratu O
